# Quantification of High-Resolution Contrast-Enhanced T1-Weighted Vessel Wall MRI for Predicting Disease Progression in Moyamoya Disease

**DOI:** 10.3390/brainsci15101089

**Published:** 2025-10-09

**Authors:** Kateryna Goloshchapova, Patrick Haas, Daniel Vogl, Lucas Wiggenhauser, Helene Hurth, Florian Hennersdorf, Benjamin Bender, Till-Karsten Hauser, Marcos Tatagiba, Nadia Khan, Constantin Roder

**Affiliations:** 1Department of Neurosurgery and Center for Moyamoya and Cerebral Revascularization, 72076 Tübingen, Germany; kateryna.goloshchapova@med.uni-tuebingen.de (K.G.);; 2Department of Diagnostic and Interventional Neuroradiology, University of Tübingen, 72016 Tübingen, Germany; 3Moyamoya Center, University Children’s Hospital Zürich and University of Zürich, 8008 Zurich, Switzerland

**Keywords:** moyamoya, vessel wall contrast-enhancement, high-resolution MRI

## Abstract

**Objective****:** In moyamoya disease (MMD), the internal carotid and proximal cerebral arteries narrow, potentially leading to stroke or hemorrhage from fragile collaterals. Disease activity and progression may be detected by contrast-enhanced (CE) high-resolution (HR) vessel wall imaging (CE-VWI) on T1-weighted MRI. However, this imaging approach needs standardization for the evaluation of signal intensity and longitudinal reproducibility. **Methods:** MMD patients with at least two separate CE-VWI examinations on the same and on different scanners were included. Signal intensity of the vessel wall, pituitary stalk, and temporal lobe white matter were measured and normalized using manually selected regions of interest. Intraindividual longitudinal reproducibility of MRI was analyzed and the clinical course was correlated with vessel wall enhancement data. **Results:** Eighty-seven patients were analyzed. Primary analysis included 60 patients with two or more CE-VWI measurements (n = 129) with median 14.8 months between examinations (range: 2–36 months) on the same scanner. Intraindividual variation in pituitary stalk enhancement (positive control) and temporal lobe white matter enhancement (negative control) showed median signal variability of 20.5% and 17.5%, respectively. The pituitary-to-temporal lobe signal intensity ratio remained stable over time (*p* = 0.843) with 9.4% median variability. Correlation analysis revealed a significant positive association between pituitary and temporal lobe signal changes (ρ = 0.717, *p* < 0.001). A total of 75% of patients showed vessel wall contrast enhancement with fluctuating signal intensity over approximately 15.9 months, likely depicting disease activity. **Conclusions:** CE-VWI is important for screening disease activity in moyamoya patients. Our findings demonstrate longitudinal intraindividual reproducibility when normalized to pituitary stalk, enabling quantified evaluation of disease progression through longitudinal vessel wall contrast-enhancement changes.

## 1. Introduction

MMD is characterized by the narrowing and occlusion of the distal internal carotid and the proximal sections of the anterior, middle, and posterior cerebral arteries [1]. The most common surgical treatment available is revascularization by direct or indirect extracranial–intracranial bypass [2].

The primary diagnostic tools for the evaluation of the disease, blood-flow and to assess disease activity are digital subtraction angiography (DSA), functional MRI and PET/CT [3,4]. Furthermore, contrast-enhanced vessel wall imaging (CE-VWI) has become a routine sequence within the last years, likely depicting disease activity and subsequent progression of the stenosis within longitudinal follow-up [5,6,7]. Since the pituitary stalk is often used as a reference measurement with maximum CE to build a quotient [5,8,9], there is an unmet need to evaluate its intraindividual signal stability over several measurements. As negative control, we chose the temporal lobe white matter in direct proximity to the pituitary stalk, an area without any contrast enhancement. Our analysis aimed to evaluate the intraindividual longitudinal signal stability and reproducibility with direct measurements of signal intensities of the pituitary stalk and the temporal lobe white matter. Based on these findings, we further analyzed the chronological course of possible VW-CE within patients normalized to the before-mentioned reference.

## 2. Methods

### 2.1. Study Design and Patient Cohort

This retrospective cohort study included consecutive adult patients with moyamoya angiopathy (MMA) who were treated at our specialized center between 2012 and 2024. Demographic data (age, sex), time of diagnosis, clinical presentation and details of surgical interventions were collected retrospectively from patient records. The main inclusion criterion was the availability of at least two CE-VWI MRI datasets on the same scanner. Grade of vascular stenosis was evaluated on DSA and correlated with subsequent MRI scans.

### 2.2. Imaging Protocol

All patients with newly diagnosed moyamoya disease undergo DSA and MRI including CE-VWI at the initial evaluation, and at the one-year follow-up. For cases involving potentially affected but untreated vascular territories, repeat DSA and/or HR-VWI are performed to assess progression or the onset of VW-CE.

An analysis of all imaging modalities (MRI, DSA) was performed for all timepoints. HR-MRI was performed on a 3 T MRI with a 64 Channel Head/Neck Coil (Siemens Healthineers AG, Forchheim, Germany). The sequence parameters were as follows: 3-dimensional T1 Sampling Perfection with Application optimized Contrasts using different flip angles, Evolution time of repetition 600 ms, time of echo 23 ms, flip angle 120°, slice thickness 0.8, field of view 185 × 220 mm, time of acquisition (TA) 3:45, Matrix 384 × 324, and resulting voxel size 0.29 × 0.29 × 0.80 mm. Contrast agent (dosage: 0.1 mmol Gadobutrol/kg) was administered during the measurement of a perfusion sequence (TA 2:29). The vessel wall imaging sequence was started after a whole-brain T1 sequence (TA 1:46) so that VW imaging was started around 4:25 min after contrast administration. Blood suppression was achieved by inversion recovery, and no pulse gating was used.

### 2.3. Measurements

The signal intensity was measured at the pituitary stalk and the temporal lobe white matter by manually selected regions of interest (ROI) on the same MRI with axial slices. If the vessel wall showed contrast enhancement, this was also measured by a manually selected ROI including the area of highest signal intensity. The maximum enhancement intensity values were selected for further quantitative evaluation within each ROI. Quotients were calculated to reference signal intensity of contrast enhancement to non-enhancing tissue (
$\frac{signal\;intensity\;of\;the\;pituitary\;stalk}{signal\;intensity\;of\;the\;temporal\;lobe\;white\;matter}$) and for quantification of VW-CE in reference to the highest possible contrast enhancement at the pituitary stalk (
$\frac{signal\;intensity\;of\;the\;maximum\;vessel\;wall\;contrast\;enhancement}{signal\;intensity\;of\;the\;pituitary\;stalk}$).

DSA (6-vessel catheter angiogram) was analyzed blinded for MRI findings for the grade of the stenosis as follows: occlusion (grade 4), severe stenosis (80–99%, grade 3), moderate stenosis (50–79%, grade 2), mild stenosis (<50%, grade 1), no stenosis (grade 0), and Suzuki stage according to Suzuki and Takaku [1]. Disease progression was defined as new or progressive stenosis of the affected vessel sections.

### 2.4. Statistical Analysis

Statistical analyses were performed using IBM SPSS Statistics for Windows (Version 25.0; IBM Corp, Armonk, NY, USA) and Microsoft Excel (Version 2405; Microsoft Corp, Redmond, Washington, USA) for graphics. For the continuous data comparison such as age, an independent samples *t*-test (Welch’s *t*-test) was used. For the analysis of the categorical parameters (sex, symptom presentation, surgery type), Fisher’s exact test (frequencies < 5) was used. For symptoms comparisons within two groups, the Bonferroni–Holm correction was used. For the comparison of Suzuki grade between patients with and without contrast enhancement, Suzuki grade was treated as an ordinal (ranked) variable and analyzed using a Mann–Whitney U (MWU) test. For the normality testing of the pituitary and temporal lobe changes, Shapiro–Wilk statistics was used. Continuous data with normal distributions were analyzed using a one-sample Student’s *t*-test with Cohen’s d for the size effect [10], while non-normally distributed data were assessed with the Mann–Whitney U test. For the correlation analysis, the non-parametric method as Spearman rank correlation was used.

Test–retest reliability was assessed using the intraclass correlation coefficient ICC(3,1) with absolute agreement, employing a two-way mixed-effects model where the single rater represents a fixed effect while subjects represent a random sample from the population of interest. Absolute agreement was chosen to capture both systematic differences and random measurement error between repeated assessments. The ‘single measures’ ICC(3,1) reflects the reliability of individual measurements as used in clinical practice, while ‘average measures’ ICC(3,1) represents theoretical reliability if multiple measurements were averaged. Cicchetti (1994) guidelines were used for ICC interpretation: values < 0.40 indicate poor reliability, 0.40–0.59 fair, 0.60–0.74 good, and 0.75–1.00 excellent reliability [11]. A two-tailed significance threshold was set at *p* < 0.05. The manuscript adheres to STROBE guidelines.

## 3. Results

### 3.1. Patient Population and Characteristics

The 219 CE-VWI sequences of 87 patients with at least 2 MR timepoints were analyzed for this study. Clinical baseline data can be found in Table 1. A total of 33% of patients had unilateral moyamoya disease upon the primary evaluation, whereas 27.6% were male. The mean age was 42.3 ± 14.0 years on primary evaluation. In total, 78/87 of patients (89.7%) underwent surgery either with direct or indirect bypass, among them 44/87 of patients (50.5%) unilaterally and 34/87 (39.1%) bilaterally. A total of 9 out of 87 patients (10.3%) did not undergo surgery at any point during the entire follow-up period. Both groups (with and without VW-CE) did not significantly differ in terms of age (43.0 ± 14.2 yo in the CE group and 39.3 ± 13.6 yo in the non-CE group, *p* = 0.287) and sex (*p* = 0.57). Analysis of initial symptoms revealed that stroke as a presenting symptom was significantly more frequent in patients with VW-CE (OR 6.0, 95% CI: 1.60–22.50, *p* = 0.021). Comparing Suzuki Grades at presentation, patients with VW-CE presented with a median grade of 3 (Inter-quartile rang (IQR) 2–4), patients without VW-CE between 3 and 4 (IQR 3–5). Therefore, patients without VW-CE had a significantly higher Suzuki grade at presentation (MWU, *p* = 0.025). When comparing the bypass status of the two groups, there was no significant difference between groups in terms of overall surgery frequencies (Fisher exact, OR 1.94, 95% CI: 0.44–8.60, *p* = 0.403). When comparing the number of territories revascularized (for example patients with both MCA and both ACA territories revascularized), no significant difference occurred between groups (Pearson Chi-square, χ^2^ = 2.88, df = 1, *p* = 0.090, OR 1.5, 95% CI: 0.94–2.39).

### 3.2. Signal Intensity of Pituitary Stalk and White Matter of the Temporal Lobe

All studies were conducted over several years using two different MRI machines, one of which underwent a software upgrade. For the initial analysis of the signal intensity of the pituitary stalk and the white matter of the temporal lobe, we included patients who had at least two scans performed on the same MRI scanner with the same preset (n = 60 patients). Then, we included all patients who had at least two MRI scans at different timepoints, including analysis on different MRI machines (n = 87 patients). The methodology for assessing pituitary stalk, temporal lobe, and vessel wall enhancement is illustrated in Figure 1.

### 3.3. Analysis of MRI Performed on the Same MR Machine

In this cohort of 60 patients, a total of 129 MRI scans with T1 SPACE sequences performed on the same MRI machine were analyzed, with a median interval of 14.8 months between examinations (range: 2–36 months).

Comparison of maximum contrast enhancement of the pituitary stalk between the two timepoints revealed a median intraindividual signal change of 20.5%, with no statistically significant difference between the two measurements (paired *t*-test, *p* = 0.058). The temporal lobe white matter showed a median intraindividual signal change of 17.5%, with a statistically significant difference between the two MRI timepoints (paired *t*-test, *p* = 0.024), but the effect size was small (Cohen’s d = 0.298). However, strong correlations were observed between changes in pituitary and temporal lobe measurements at both timepoints: baseline measurements showed robust correlations (Spearman ρ = 0.815, *p* < 0.001) for the temporal–pituitary correlation at the first MRI timepoint, while follow-up measurements maintained these significant associations (Spearman ρ = 0.780, *p* < 0.001) for the second MRI timepoint. Changes in absolute signal intensities of the pituitary stalk and temporal lobe consistently occurred in the same direction within individuals. Consequently, the ratio of pituitary stalk-to-temporal lobe signal intensity remained stable over time, with a mean intraindividual change of 9.4%, with no statistically significant difference between the two measurements (paired *t*-test, *p* = 0.843; see Figure 2). The correlation analysis between pituitary and temporal lobe signal changes revealed a significant positive association with a strong monotonic relationship (ρ = 0.717, *p* < 0.001), accounting for 50.9% of the variance. To evaluate the consistency (reliability) between the two MRI scans, the intraclass correlation coefficient (ICC(3,1)) was calculated. For the pituitary stalk, the ICC(3,1) for single measures was 0.468 (95% CI: 0.248–0.642; F = 2.836, *p* < 0.001), with average measures of 0.637 (95% CI: 0.398–0.782). For the temporal lobe white matter, the ICC(3,1) was 0.434 (95% CI: 0.209–0.617; F = 2.646, *p* < 0.001), with average measures of 0.605 (95% CI: 0.345–0.763). According to the established criteria, these values indicate reliability for the test–retest evaluation of pituitary stalk and temporal lobe white matter signal intensities.

### 3.4. Analysis of MRI Performed on Different MR Machines

Analysis of all patients with at least two MRI timepoints (n = 87 patients) revealed a median intraindividual signal change of 33.5%, with no statistically significant difference between the two measurements (paired *t*-test, *p* < 0.001) for pituitary stalk and of 24.7% (paired *t*-test, *p* < 0.001) for temporal lobe white matter. The correlations between pituitary and temporal lobe measurements at both MRI timepoints remained strong (Spearman ρ = 0.787, *p* < 0.001 for the first MRI timepoint, and Spearman ρ = 0.803, *p* < 0.001 for the second MRI timepoint). Therefore, stable results were seen for the pituitary stalk/temporal lobe quotient, with a mean change of 9.6%, with no statistically significant difference between the two measurements (paired *t*-test, *p* = 0.843) (Figure 3). Correlation analysis between pituitary and temporal lobe signal changes again revealed strong agreement (ρ = 0.830, *p* < 0.001).

### 3.5. Vessel Wall CE Dynamics

Based on these findings showing strong stability of the intensity signal, all further analyses were normalized based on the maximum intensity value of the pituitary stalk for normalization.

In the entire cohort (87 patients), vessel wall contrast enhancement (VW-CE) was observed in 75% of patients on at least one imaging study. The longitudinal analysis of VW-CE showed dynamic changes over time with an initial increase and a secondary decrease after reaching the highest intensity values.

To further assess these longitudinal dynamics of VW-CE, we separately analyzed patients with a minimum of three MRI scans (n = 23 patients, VW-CE on the right side n = 19, VW-CE on the left side n = 18, bilateral VW-CE n = 14). Each hemisphere was analyzed independently. Figure 4 illustrates the assessment of VW-CE for one representative patient.

For each patient, the timepoint with the highest VW-CE-to-pituitary stalk signal ratio (used as a positive control) was designated as t = 0. Polynomial curve fitting was applied to the available timepoints and corresponding ratios (Figure 5). The resulting curve formed an inverted parabola, supporting the observed pattern of both increasing and decreasing VW-CE intensity over time (R^2^ = 0.081 for quadratic polynomial curve with F = 6.1 with *p* = 0.03).

### 3.6. Correlation with Stroke, CVR, and Angiographic Stenosis

In addition to MRI, we analyzed 105 DSA for quantification and change in the stenosis of each vessel. We specifically analyzed patients presenting with mild to moderate stenosis of the terminal internal carotid artery (ICA) or its proximal branches, tracking the chronological relationship between stenosis progression and changes in VW-CE (see Figure 6). Patients who already presented with complete ICA occlusion were excluded, as no progression can be depicted in these patients. A total of 18 patients with 70 MRI and 50 DSA timepoints were analyzed. We were able to depict an increase in stenosis seen on DSA in correlation with the change in the intensity of CE on VWI especially towards the maximum vessel wall contrast uptake.

## 4. Discussion

To our knowledge, no standardized methodology has been established to verify the stability and comparability of MR signal intensity in VWI, a prerequisite for valid longitudinal comparisons. Several studies have employed reference tissues such as the pituitary stalk, adjacent gray matter, temporal muscle, or pre-contrast imaging of the same tissue, predominantly utilizing T1-weighted sequences [12,13]. The longitudinal reproducibility of VW-CE has not been thoroughly addressed in previously reported studies. With this study, we demonstrated fair reproducibility of signal intensity scores for the pituitary stalk (positive control) and temporal lobe white matter (negative control), which enabled further quantitative evaluation of VW-CE. Further, we were able to show an increasing and decreasing VW-CE intensity during the course of the disease, which correlated with angiographic disease progression [6,7].

The first semi-quantitative method for evaluating contrast enhancement was described by Wang et al. [5]. Subsequent studies identified the lower portion of the infundibulum as a reliable reference structure for assessing signal intensity on post-contrast imaging, demonstrating the highest interrater agreement (0.7–0.9) among all examined tissues [8,9,14]. “High-grade” VW-CE was defined by an intensity quotient ranging from 0.8 to 1.0 relative to the pituitary stalk. Conversely, the definition of “no or minimal contrast enhancement” relied on visual comparison with non-contrast images or an intensity value below 20% compared to the pituitary stalk [5]. However, these studies exclusively assessed signal intensity on single MRI scans and did not evaluate the longitudinal stability of this method.

With our data we were able to show that the longitudinal use of CE-VWI for the evaluation of disease activity is reliable and reproducible on the same and on different MR scanners. We were able to show that the mean intraindividual variation in the pituitary stalk (as positive control) and parenchyma of the temporal lobe (as negative control) enhancement is approximately 20% with shifts consistently occurring in the same direction. When considering the more relevant quotient between white matter and the pituitary stalk, the signal variation is lower, at 9.4%.

Most studies report a similar proportion of patients exhibiting vessel wall contrast-enhancement (VW-CE), typically ranging from 70% to 90% [15,16]. However, several other studies document significantly lower rates, observing VW-CE in approximately 30% and 50% of their cohorts [8,17,18]. A key factor likely contributing to this discrepancy is the exclusion of patients with unilateral or bilateral complete occlusion of the internal carotid artery (ICA) in those studies, based on evaluations using digital subtraction angiography (DSA). In contrast, in our cohort, 67.3% of patients already exhibited occlusion of the distal ICA on at least one side during the initial assessment with conventional angiography. Furthermore, studies report varying temporal dynamics of contrast enhancement progression, ranging from 7.8% as documented by Lu et al. to 42.3% observed by Muraoka et al., which may reflect differences in disease activity among distinct patient populations [8,17]. Our findings regarding symptomatology also differ substantially from those reported in previous cohorts. Lim et al. observed that 81% of 19,700 patients were asymptomatic, whereas in our cohort more than 80% of patients presented with symptoms at the time of diagnosis, which may represent different diagnostic approaches [19]. Studies with different ethnicities also introduce variations in the incidence of MMD as well as in its predominant clinical presentation (sex and age distribution, percentage of hemorrhagic vs. ischemic presentation) [20]. Differences in disease manifestations and timing of imaging (i.e., CE-VWI after initial symptoms as according to our protocol versus CE-VWI at any later point of the disease as the screening of patients) might therefore play a crucial role in detecting comparable fractions of patients with and without VW-CE.

Overall, the variability in vessel wall contrast-enhancement rates among moyamoya patients reflects heterogeneous cohorts comprising individuals at different disease stages, resulting from diverse local protocols governing patient evaluation, diagnosis, treatment, and follow-up.

Our study corroborates findings from other investigators, demonstrating that patients exhibiting contrast enhancement display distinct immediate disease trajectories and progression dynamics compared to those without contrast enhancement. This is reflected in different clinical presentations, including elevated stroke risk (OR 6.0 in our cohort), as well as varying disease stages as evidenced by Suzuki staging (higher Suzuki stages observed in patients without contrast enhancement). Other studies demonstrated a correlation between the clinical and angiographical progression of MMD and the intensity of VW-CE [5,6,7,8,21,22]. Moreover, it was shown that not only the presence or absence of VW-CE was relevant for prognosis, but also its intensity and longitudinal changes over time [5,8,22].

One of the earliest investigations into the relationship between stroke and vessel wall contrast-enhancement (VW-CE) in moyamoya disease (MMD), conducted by Ryoo et al. (2014), did not demonstrate a correlation between vessel wall enhancement and symptom progression [23]. This absence of association may be attributable to the limited sample size (n = 32) and the reliance on a single-point VW-CE assessment, rather than longitudinal follow-up to evaluate disease progression.

Regarding the dynamics of vessel wall contrast-enhancement (VW-CE), Roder et al. and Wang et al. have demonstrated that VW-CE increase and decrease over time [6,7]. Lu et al. also described longitudinal changes in VW-CE [8]; however, their study concluded that VW-CE is stable and persistent, likely due to their relatively coarse classification of enhancement intensity—categorized only as absent, less than, or greater than pituitary stalk enhancement—and a comparatively short follow-up period (10 months in Lu et al., 2023) versus the longer 34.6 months in our sub-cohort of 23 patients, which included a separate evaluation of both arterial sides. By contrast, our findings indicate that VW-CE intensity initially increases then decreases across the course of disease progression, while its peak might have a temporal correlation to the progression of vessel stenosis in the affected artery.

Beyond conventional and invasive angiography, various diagnostic modalities have emerged to assess moyamoya disease progression, both ipsilaterally and contralaterally. Hemodynamic assessment techniques, which complement morphological studies, encompass CO_2_-triggered BOLD MRI [24], H_2_^15^O PET with acetazolamide challenge [25,26], and the recently developed CTA-based computational fluid dynamics models [27]. These hemodynamic studies serve distinct clinical purposes: while hemodynamic evaluation primarily guides changes in blood-flow and revascularization decisions, morphological studies might enable a broader understanding of the disease and possible prognostic factors. Vessel wall imaging offers additional aspects by enabling disease progression monitoring and likely facilitating differentiation from conditions such as vasculitis and atherosclerosis [28].

Despite being able to reliably evaluate VW-CE on longitudinal MRI datasets, the origin of this contrast enhancement in moyamoya patients remains unclear. VW-CE has been explained as inflammation of the vessel, cholesterol, calcium deposition in the vessel, disruption of the intima or contrast pooling in the vessel wall in other pathologies such as atherosclerosis, among others [29,30,31,32].

The suggested pathophysiology for thickening of the vessel wall in MMD is smooth muscle cell proliferation, degeneration or death, elastic lamina fragmentation, or necrotic cellular component accumulation, which leads to luminal narrowing and may explain concentric enhancement [18,33]. Moyamoya has to be differentiated from cerebral vasculitis, where patients may also show VW-CE, but in more distal vessel segments [28]. In vasculitis patients, it is thought to be attributable to neovascularization and proliferation of the vasa vasorum on the outer vessel wall and permeability of the endothelium due to inflammation and cytokines and extravasation of the contrast agent into the vessel wall [31,34]. In studies of patients with intracranial aneurysms, there was evidence that the endothelial dysfunction might contribute to VW-CE [32]. This endothelial dysfunction might not be caused by inflammation, but by mechanical stress of the vessel wall caused by turbulent blood flow. However, as the VW-CE only appears in proximal segments of the main cerebral arteries, biopsies are not feasible and therefore the pathophysiology remains unclear.

## 5. Limitations

This is a retrospective analysis with a primarily single-rater reading of the contrast-enhanced images. However, we have placed larger ROIs than the actual size of the relevant structure and took the automatically calculated highest intensity score of the entire ROI for further evaluation. Therefore, multiple reads might not have increased the meaningfulness of this analysis. The comparison of VW-CE and disease progression seen on DSA also has a small sample size since we included vessel segments with no occlusion on the initial DSA. The exact pathophysiological process causing VW-CE in moyamoya patients remains unclear. This remains a critical question to address in order to better understand the pathophysiology of moyamoya.

## 6. Conclusions

Contrast-enhanced T1 black blood sequences with high spatial resolution may serve as a reliable screening tool for moyamoya patients. At the time of initial diagnosis, this method might be able to depict disease activity and therefore might have the potential for early detection of progression, such as in the contralateral hemisphere. Given its good longitudinal reproducibility, vessel wall sequences should be routinely used as a long-term imaging marker, alongside other biomarkers, to possibly improve the understanding of moyamoya disease in the future.

## Figures and Tables

**Figure 1 brainsci-15-01089-f001:**
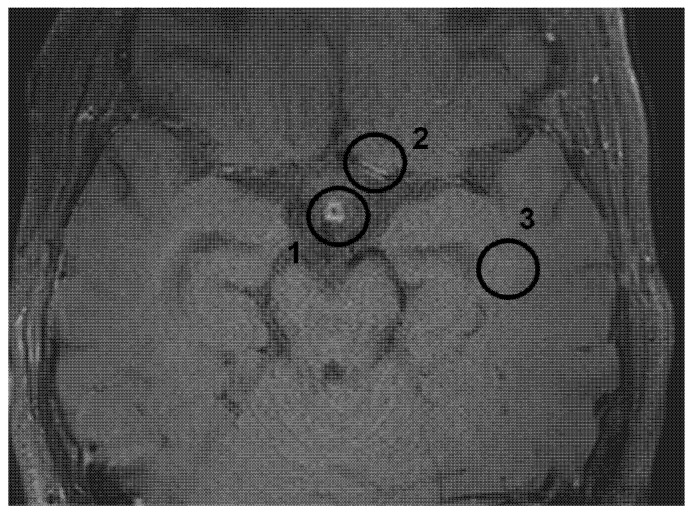
Evaluation of signal intensity of the pituitary stalk (1) and temporal lobe (3) and of the vessel wall (2). For the analysis, the automatically calculated maximum intensity value of the respective ROI was chosen.

**Figure 2 brainsci-15-01089-f002:**
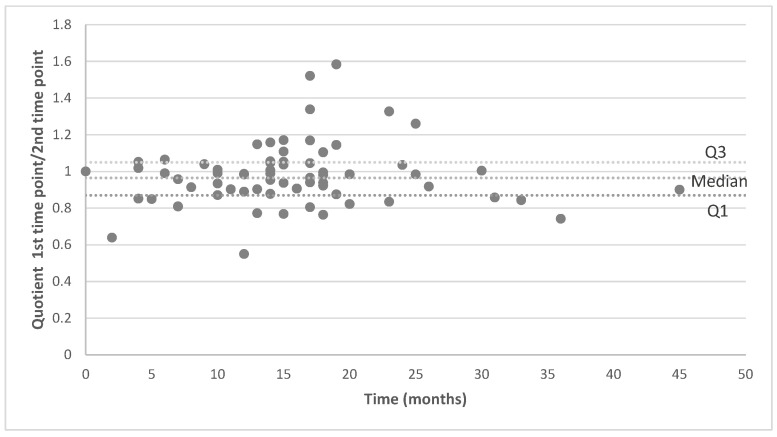
Longitudinal analysis of the comparison of the quotients (pituitary stalk/temporal lobe signal intensity) between first and second imaging timepoint for patients with MRI on the same machine only. The median signal change for 69 timepoints is 9.4%.

**Figure 3 brainsci-15-01089-f003:**
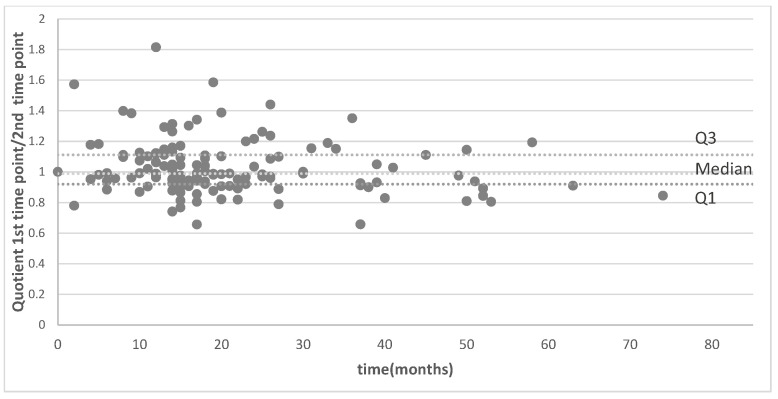
Longitudinal analysis of the comparison of the quotients (pituitary stalk/temporal lobe intensity) between first and second imaging timepoint for patients with MRI on the same machine only. The median signal change for 134 timepoints is 9.6%.

**Figure 4 brainsci-15-01089-f004:**
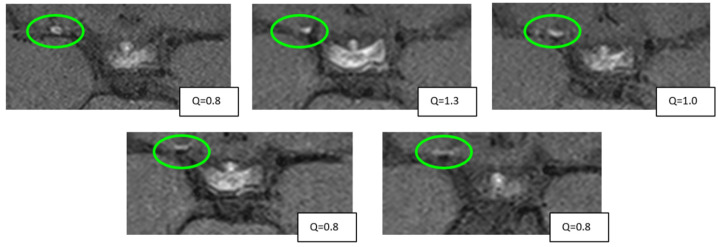
Exemplary case of the longitudinal evaluation of the VW-CE of the same patient on T1 Space sequence. The green oval indicates the region of interest (ROI) where the maximum intensity for VW-CE was measured. In the corresponding lower right corner, the ratio of intensities between VW-CE and the pituitary stalk is displayed.

**Figure 5 brainsci-15-01089-f005:**
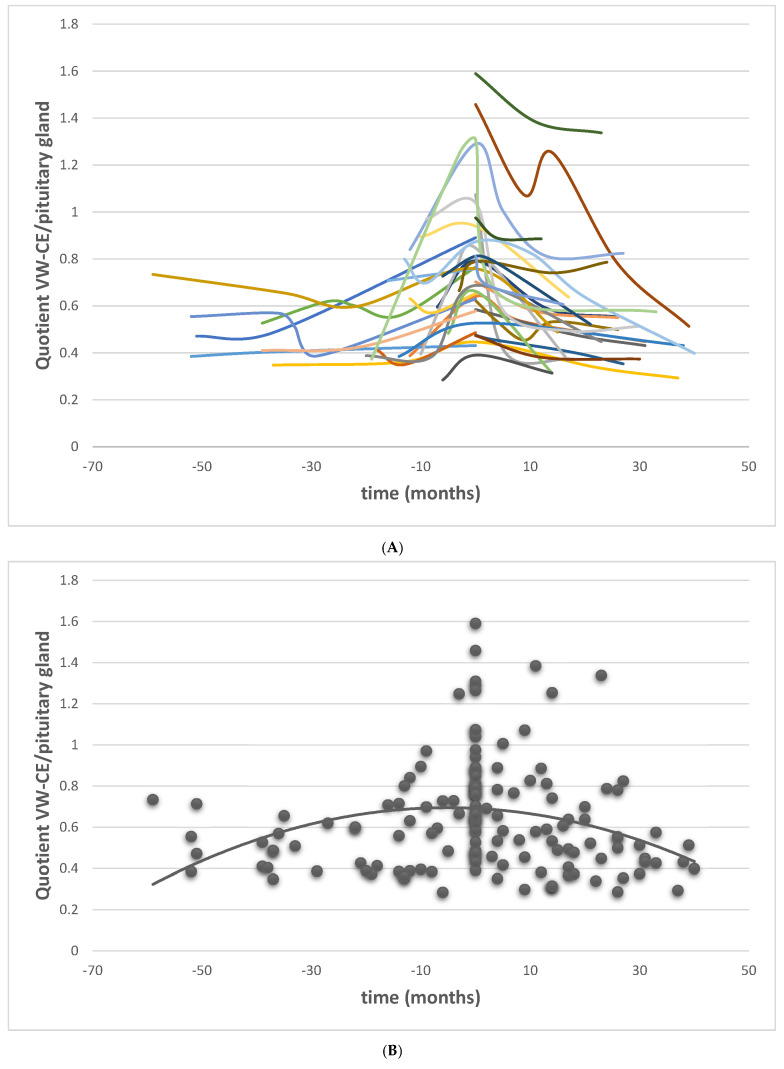
Longitudinal development of VW-CE. (**A**). Vessel wall-to-pituitary stalk ratio of 20 patients (37 curves, 19 for the right side, 18 for the left side) with at least 3 MRI scans with a total of 142 data points (MRI scans; mean 3.8 scans per patient) in chronological order. (**B**). Trend line fitted as inverse parabola to all MRI timepoint. Timepoint t = 0 months is referenced as the time of the highest VW-CE intensity.

**Figure 6 brainsci-15-01089-f006:**
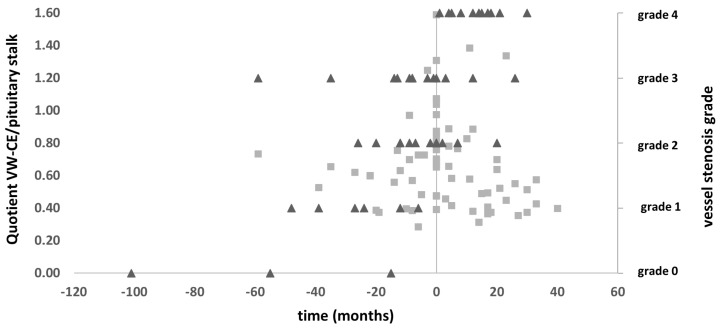
Longitudinal analysis of WV-CE and angiographic stenosis of MMD patients with initial mild to moderate stenosis (n = 18, 70 MRI datapoint and 50 DSA data points). Squares depict the datapoint from MRI (quotient VW-CE/pituitary stalk with t = 0 as the datapoint with maximal VW-CE for the individual patient). Triangles depict the level of stenosis of the corresponding vessel segment on DSA: occlusion (grade 4); severe stenosis (80–99%, grade 3), moderate stenosis (50–79%, grade 2), mild stenosis (<50%, grade 1), no stenosis (grade 0).

**Table 1 brainsci-15-01089-t001:** Patient characteristics.

	All Patients (n = 87)	With CE (n = 68)	Without CE (n = 19)	*p*-Value
**Age**, yo (± SD)	42.3 ± 14.0	43.0 ± 14.2	39.3 ± 13.6	*p* = 0.287
**Sex**				*p* = 0.57
- female	63 (72.4%)	48 (70.6%)	15 (78.9%)	
- male	24 (27.6%)	20 (29.4%)	4 (21.1%)	
**Symptoms** (on primary evaluation)				
- TIA	26 (29.9%)	21 (30.9%)	5 (26.3%)	*p* = 1.0
- Stroke	39 (44.8%)	36 (52.9%)	3 (15.8%)	***p* = 0.021 ***
- ICH	7 (8.0%)	3 (4.4%)	4 (21.1%)	*p* = 0.191
- Minor (headache, performance deficits)	11 (12.6%)	7 (10.3%)	4 (21.1%)	*p* = 1.0
- Other (incidental, syncope, epileptic seizure, positive family history, etc.)	4 (4.6%)	1 (1.5%)	3 (15.8%)	*p* = 0.157
**Suzuki grade:** right/left (sum right + left, % of all)				***p* = 0.025 ***
not affected (right/left (total, %))	13/21 (34, 19.5%)	8/19 (27, 19.9%)	5/2 (7, 18.4%)	
grade 1 (right/left (total, %))	7/7 (14, 8.5%)	6/7 (13, 9.6%)	1/0 (1, 2.6%)	
grade 2 (right/left (total, %))	14/13 (27, 15.5%)	12/11 (23, 16.9%)	2/2 (4, 10.5%)	
grade 3 (right/left (total, %))	27/21 (48, 27.6%)	26/15 (41, 30.1%)	1/6 (7, 18.4%)	
grade 4 (right/left (total, %))	14/9 (23, 13.2%)	8/5 (13, 9.6%)	6/4 (10, 26.3%)	
grade 5 (right/left (total, %))	6/9 (15, 8.6%)	5/7 (12, 8.8%)	1/2 (3, 7.9%)	
grade 6 (right/left (total, %))	6/7 (13, 7.5%)	3/4 (7, 5.1%)	3/3 (6, 15.8%)	
**Bypass status**				*p* = 0.090
- without surgery	9 (10.3%)	6 (8.8%)	3 (15.8%)	
- STA-MCA bypass				
unilateral	44 (50.6%)	35 (51.5%)	9 (47.4%)	
bilateral	34 (39.1%)	27 (39.7%)	7 (36.8%)	
- STA-ACA bypass				
unilateral	10 (11.5%)	8 (11.8%)	2 (10.5%)	
bilateral	20 (23.0%)	18 (26.4%)	2 (10.5%)	
- OA-PCA bypass				
unilateral	3 (3.4%)	3 (4.4%)	0 (0%)	
bilateral	1 (1.1%)	1 (1.5%)	0 (0%)	

Abbreviations. TIA: transient ischemic attack; ICH: intracranial hemorrhage; STA-MCA bypass: superficial temporal artery to middle cerebral artery bypass; STA-ACA bypass: superficial temporal artery to anterior cerebral artery bypass; OA-PCA bypass: occipital artery to posterior cerebral artery bypass; *: statistical significant.

## Data Availability

Anonymized data not included in this publication is available upon reasonable request to qualified investigators. The data are not publicly available due to privacy and ethical restrictions.

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
