# Peer review of "Quantification of High-Resolution Contrast-Enhanced T1-Weighted Vessel Wall MRI for Predicting Disease Progression in Moyamoya Disease"

_brainsci, 2025, doi:10.3390/brainsci15101089_

Round 1
Reviewer 1 Report
Comments and Suggestions for Authors
I appreciate the authors for presenting this clinically relevant research article, which investigates the quantification of high-resolution contrast-enhanced T1-weighted vessel-wall MRI for predicting disease progression in Moyamoya disease. This study provides an important methodological and clinical contribution by demonstrating the longitudinal reproducibility of CE-VWI using the pituitary stalk as a reference structure. The results showed that VW-CE intensity exhibits a dynamic pattern, initially increasing and subsequently decreasing with disease progression, and correlates with stenosis changes on DSA. These findings offer new insights into the pathophysiology of MMD and hold potential value for disease monitoring. Importantly, the study supports VW-CE as a dynamic biomarker of disease activity, although its underlying biological mechanisms remain to be elucidated.
Overall, this is a well-designed and well-organized study. My only suggestion is that the authors provide a more explicit comparison with previous studies reporting discrepant VW-CE frequencies (30% vs. 70–90%), and clarify how their methodology addresses and overcomes the limitations of earlier work.
Comments on the Quality of English Language
The English could be improved to more clearly express the research.
My only suggestion is that the authors provide a more explicit comparison with previous studies reporting discrepant VW-CE frequencies (30% vs. 70–90%), and clarify how their methodology addresses and overcomes the limitations of earlier work.
Author Response
My only suggestion is that the authors provide a more explicit comparison with previous studies reporting discrepant VW-CE frequencies (30% vs. 70–90%), and clarify how their methodology addresses and overcomes the limitations of earlier work.
Thank you very much for the comment. We believe that the main difference of the varying frequencies of vessel wall contrast enhancement observed at different patient cohorts might be explained with different imaging protocols and acute disease stages at the time-point of imaging. For example, we perform vessel wall imaging in every patient with a suspicion of Moyamoya Disease and therefore do not miss a patient with contrast enhancement. Since contrast enhancement is dynamic and mainly seen in early stages, it is only rarely seen in more chronic patients with higher Suzuki grades. The discussion in the manuscript has been revised accordingly. (Page 10)
Reviewer 2 Report
Comments and Suggestions for Authors
The early detection of MMD and its progression is an important clinical need. This paper enriched the research via radiological approach. However, I have some concerns on the methodology.
- The sample size is small, with unbalanced distribution between two subgroups, compromising the reliability of the conclusion.
- In Figure 5 there is no quantitative analysis (e.g., adjusted R-squared). The samples are not distributed evenly. The high sample density at 0 might distort the results.
- The conclusion extended beyond the observation results.
Author Response
1. The sample size is small, with unbalanced distribution between two subgroups, compromising the reliability of the conclusion.
We are not entirely certain which of the two subgroups you are referring to. Perhaps you mean that, for the longitudinal analysis, we only included patients who had not yet developed a complete vessel occlusion? In that case, the subgroups are distributed quite similarly (18 out of 37 patients, 48.6%, with at least three HR-VWI examinations were analyzed). The relatively small patient number is mainly due to the fact, that only patients with at least three MRI scans were included. The total study cohort comprised 87 patients with at least two MRI scans. Altogether, the number of patients with moyamoya at our center who underwent at least one MRI amounts to 218. If you refer to the disbalance between patients with and without contrast enhancement, we think that this should not compromise the conclusion of this paper, as the main aim was to depict signal stability between pituitary stalk and white matter signal intensities.
2. In Figure 5 there is no quantitative analysis (e.g., adjusted R-squared). The samples are not distributed evenly. The high sample density at 0 might distort the results.
Thank you for this remark. We have now revised the manuscript to include the data with the corresponding R² value. In addition, we performed a mixed F-test for the distribution. The polynomial function yielded the best R² values with statistical significance compared, for instance, to a linear regression (p-values, F-values, and adjusted R²), although the explained variance remains relatively low at around 8–10%. It is correct that the largest number of data points is at x = 0; however, this results from the definition of the zero point (x = 0) in which the ratio of VW-CE to pituitary stalk enhancement reaches its maximum. (Page 7)
3. The conclusion extended beyond the observation results.
Thank you for your careful comment regarding our correlation results on disease progression in relation to the degree of stenosis assessed by DSA and the corresponding vessel wall contrast enhancement.
You are correct that our data cannot prove the claim that the peak vessel wall contrast enhancement always precedes the development of vascular stenosis, especially since patients who already presented with complete occlusion at baseline were excluded. However, our data can at least graphically illustrate (that is why we used the term “depict”) that this hypothesis may hold true. When fitting a saturation curve to the progression of vascular stenosis (equation: y = 1.115 / (1 + e^(-0.093(t+2.80))) + 0.491; R² ≈ 0.492; p < 0.001), we obtain an inflection point at t₀ = –2.80 months, corresponding to the time of the steepest slope in stenosis development. This point is temporaly almost identical to the maximum contrast enhancement (which, as defined in our paper, is set to t = 0).
In line with this, we have modified the wording in the Results section by replacing “confirming” with “supporting” in order to appropriately soften our conclusions. (Page 7)
Reviewer 3 Report
Comments and Suggestions for Authors
This submission reports on a single center retrospective study of standardising findings in areas of the brain used as control areas, so as to allow assessment disease activity of moya-moya patients assessed by contrast-enhanced (CE) high-resolution (HR) vessel wall imaging (CE-VWI) on T1-weighted MRI. The authors found that there was intraindividual variation on pituitary stalk enhancement and temporal lobe white matter enhancement, but the pituitary-to-temporal lobe signal intensity ratio was stable over time and positively correlated. This was despite 75% of patients showing VWCE with fluctuating intensity over time that was likely related to disease activity. This supports the practice of normalising CE-VWI to the pituitary stalk.
There are some issues the authors may wish to address. Identifying the exact area of concern is made difficult by the lack of line numbering in the paper
- In line with current practice, please mention ‘female’ data in preference to ‘male’, or before male data eg Results line 3, Table 1
- Say ‘sex’ instead of ‘gender’ unless it was gender that was assessed instead of biological sex eg Results line 9. Table 1
- Missing units eg Results line 8
- Table 1 - suggest align column 1 to the left; needs a legend to explain the acronyms
- Discussion (major) - start with a qualitative summary of the main study findings. Subsequent paragraphs should then explore each main finding and compare with other studies
- Discussion - para 1-3 – better absorbed into the Introduction
- Discussion - did surgery affect the findings?
Author Response
- In line with current practice, please mention ‘female’ data in preference to ‘male’, or before male data eg Results line 3, Table 1.
We have accordingly changed the wording
- Say ‘sex’ instead of ‘gender’ unless it was gender that was assessed instead of biological sex eg Results line 9. Table 1
We have accordingly changed the wording.
- Missing units eg Results line 8
We have added the missing units.
- Table 1 - suggest align column 1 to the left; needs a legend to explain the acronyms
We have added the abbreviations to table 1.
- Discussion (major) - start with a qualitative summary of the main study findings. Subsequent paragraphs should then explore each main finding and compare with other studies
- Discussion - para 1-3 – better absorbed into the Introduction
Thank you for the remarks 5.) and 6.). The discussion has been revised accordingly. As you suggested, we shortened paragraphs 1–3 and reorganized the remaining sections.
- Discussion - did surgery affect the findings?
Regarding the response to Question 7, we cannot make a reliable statement about the influence of bypass surgery on vessel wall contrast enhancement, since only 3 out of 215 patients in our total cohort were not operated, whereas the remaining patients underwent one or more bypass procedures. However, from our clinical experience we can state that patients after bypass surgery usually show no relevant changes of contrast enhancement besides the before-mentioned in- and decreasing pattern over time. However, we currently run a prospective study, including this topic with longitudinal vessel wall imaging in even closer time-frames than the data of this study.
Round 2
Reviewer 2 Report
Comments and Suggestions for Authors
Thanks for the update. Some major concerns have been addressed. I have a couple of suggestions for further improvement.
- The unbalance between the between patients with and without contrast enhancement deserves further discussion. And I suggest to further tone done the conclusion considering the sample size.
- I suggest the discuss the results in a comparative perspective. It is well known that MMD patients go through vascular reshaping, leading to the progression even after treatment. Some computational models have been developed to evaluate its hemodynamic influence (ref: 10.1007/s00701-022-05455-9, 10.1038/s41598-024-79608-4). A comprehensive, in-depth comparison between imaging and computational approaches can enhance the understanding of the advantages of the proposed method.
Author Response
1. The unbalance between the patients with and without contrast enhancement deserves further discussion. And I suggest to further tone done the conclusion considering the sample size.
We have attempted to address the variable distribution of patients with and without VW-CE comparing it to other papers, which most likely relates to the different disease stages (some VW-CE studies excluded patients with ICA occlusion at the time of diagnosis), varying imaging protocols, and different patient cohorts (for example, different imaging time-points in patient with recent symptoms versus screening examinations with patients at any later stages). We also discussed the role of different follow-up times, which may underestimate VW-CE dynamics. We have expanded the discussion section (page 10).
Concerning the second part of the reviewer’s comment: We have now changed our wording in the conclusions section accordingly to tone down our conclusions.
2. I suggest the discuss the results in a comparative perspective. It is well known that MMD patients go through vascular reshaping, leading to the progression even after treatment. Some computational models have been developed to evaluate its hemodynamic influence (ref: 10.1007/s00701-022-05455-9, 10.1038/s41598-024-79608-4). A comprehensive, in-depth comparison between imaging and computational approaches can enhance the understanding of the advantages of the proposed method.
Thank you for this suggestion. We have added a paragraph to the discussion (page 11).
Reviewer 3 Report
Comments and Suggestions for Authors
This is a revised submission of a report on a single center retrospective study of standardising findings in areas of the brain used as control areas, so as to allow assessment disease activity of moya-moya patients assessed by contrast-enhanced (CE) high-resolution (HR) vessel wall imaging (CE-VWI) on T1-weighted MRI.
I thank the authors for addressing most of my concerns. What is still not done is:
‘Discussion (major) - start with a qualitative summary of the main study findings.’
Author Response
Thank you for that comment. We have addressed this at the beginning of the discussion (page 9).
Round 3
Reviewer 2 Report
Comments and Suggestions for Authors
Thanks for the update. My previous comments have been addressed.